# SpAD Biofunctionalized Cellulose Acetate Scaffolds Inhibit *Staphylococcus aureus* Adherence in a Coordinating Function with the von Willebrand A1 Domain (vWF A1)

**DOI:** 10.3390/jfb13010021

**Published:** 2022-02-21

**Authors:** Stefanos Pendas, Antonis Asiminas, Alexandros Katranidis, Costas Tsioptsias, Maria Pitou, Georgios Papadopoulos, Theodora Choli-Papadopoulou

**Affiliations:** 1Laboratory of Biochemistry, Department of Chemistry, Aristotle University of Thessaloniki, 54124 Thessaloniki, Greece; spidy1927@yahoo.gr (S.P.); margeopit@chem.auth.gr (M.P.); 2General Hospital of Thessaloniki “St. Demetrios”, Elenis Zografou 2, 54634 Thessaloniki, Greece; 3Division of Glial Disease and Therapeutics, Center for Translational Neuromedicine, University of Copenhagen, Nørre Alle 14, 24.3.9, 2200 Copenhagen N, Denmark; a.asiminas@sund.ku.dk; 4Institute of Biological Information Processing IBI-6, Forschungszentrum Jülich (FZJ), 52425 Jülich, Germany; a.katranidis@fz-juelich.de; 5Department of Chemical Εngineering, Aristotle University of Thessaloniki, 54124 Thessaloniki, Greece; ktsiopts@gmail.com; 6Department of Biochemistry and Biotechnology, University of Thessaly, Biopolis, 41500 Larissa, Greece; geopap@bio.uth.gr

**Keywords:** cellulose acetate scaffolds, *S. aureus* protein A, von Willebrand factor A1, antibacterial properties

## Abstract

*Staphylococcus aureus* is one of the major pathogens causing and spreading hospital acquired infections. Since it is highly resistant to new generation antibiotics, novel strategies have to be developed such as the construction of biofunctionalized non-adherent surfaces that will prevent its tethering and subsequent spread in the hospital environment. In this frame, the domain D of protein A (SpAD) of *S. aureus* has been immobilized onto cellulose acetate scaffolds by using the streptavidin/biotin interaction, in order to study its interaction with the A1 domain of von Willebrand factor (vWF A1), a protein essential for hemostasis, found in human plasma. Subsequently, the biofunctionalized cellulose acetate scaffolds were incubated with *S. aureus* in the presence and absence of vWF A1 at different time periods and their potential to inhibit *S. aureus* growth was studied with scanning electron microscopy (SEM). The SpAD biofunctionalized scaffolds perceptibly ameliorated the non-adherent properties of the material, and in particular, the interaction between SpAD and vWF A1 effectively inhibited the growth of *S. aureus*. Thus, the exhibition of significant non-adherent properties of scaffolds addresses their potential use for covering medical equipment, implants, and stents.

## 1. Introduction

Hospital acquired infections caused by microorganisms such as bacteria and fungi are of great concern because of the detrimental effects caused by sepsis. Among the most common, *Staphylococcus aureus* is highly associated with hospital acquired infection. In the last years, the rapidly increasing resistance of *S. aureus* to new generation antibiotics such as linezolid [1,2] and teicoplanin as well as the limited input of new drugs with antibacterial activity has made highly imperative the need to develop novel strategies to confront this contagious pathogen.

One such strategy is the construction of non-adherent surfaces that will inhibit the in situ tethering of *S. aureus* on stents, implants, and surgical tools and its further spread in the hospital environment. This can be achieved by covering medical equipment and clothing with appropriately modified biofunctionalized materials. In the last decades, the modification and surface treatment of nanofibrous scaffolds has been of great interest. The nanofibrous scaffolds are constructs with a high surface to volume ratio, consisting of interwoven polymer fibers 100 nm to 1 μm in diameter. Their surface can be physically or chemically treated so that biomolecules with the desired properties can be tethered on them [3]. Biopolymeric scaffolds enriched with antibiotics have already been used to construct antibacterial surfaces [4,5].

The technique of electrospinning has been applied to various polymers [6,7,8,9], polymer composites [10,11,12], or polymers with encapsulated substances [13,14,15,16]. Electrospinning of cellulose acetate has been extensively studied for biomedical or other applications [9,14,15,16,17,18,19,20]. Cellulose acetate is a well-known derivative of cellulose, the most abundant polysaccharide on earth. In contrast to cellulose, which exhibits very poor solubility, cellulose acetate is soluble in many common organic solvents [17,18], and thus has been widely used in many applications. Amoxicillin is a common penicillin-like antibiotic drug that is being used in the treatment of infections such as bronchitis, pneumonia, and skin infections [20]. It has been encapsulated into biodegradable polymer micro-spheres for controlled release by different methods [20,21]. The controlled release of a drug limits the danger of overdosing.

As aforementioned, *S. aureus* is a common cause of hospital acquired infections. It is able to colonize a great variety of sites on the human body through the interaction between specific bacterial determinants and host-cell proteins. Among these determinants, proteins involved in cell wall biosynthesis, surface dynamics, and immunological response are of great interest. Staphylococcal protein A (SpA) is a major virulence protein that enables *S. aureus* invasion to the host cells through modulating the host’s immune system. This protein is a surface protein belonging to the microbial surface components and recognizes adhesive matrix molecules (MSCRAMMs). It attaches to the cell wall via its C-terminal domain and interacts with the immunoglobulin IgG via the N-terminal domains E, D, A, B, and C [22,23]. SpA is also known to bind the A1 domain of von Willebrand factor (vWF A1), a protein essential for hemostasis, found in human plasma [23].

Based on this interaction, we hypothesized that the interaction of vWF A1 with SpA could potentially inhibit the growth of *S. aureus.* To further examine this, the interaction between SpAD and vWF A1 was in silico evaluated with molecular simulation experiments. Additionally, cellulose acetate (CA) scaffolds were biofunctionalized with the domain D of protein A (SpAD) and their further interaction with *S. aureus* was investigated with or without the addition of vWF A1.

## 2. Materials and Methods

### 2.1. Materials

The pGEX-KG-SpAD vector containing the gene of SpAD was kindly provided by the Department of Microbiology of Moyne Institute of Preventive Medicine at Trinity College (Dublin, Ireland). The pAN5 plasmid for *in vivo* biotinylation of SpAD was purchased from Avidity (Denver, CO, USA). The primers for the PCR reactions were produced by MWG (Ebersberg, Germany) and enzymes were purchased from Biolabs (Hitchin, UK). The protein has been expressed in *E. coli* and further purified using Ni-NTA agarose columns from Qiagen (Hilden, Germany). Polyethylene glycol mPEG-NHS ester and the biotinylated derivative for the biofunctionalization of scaffolds was purchased from Nanocs (New York, NY, USA). All chemical reagents were purchased from Sigma-Aldrich (St. Louis, MO, USA) and were reagent grade. Millipore water was used throughout this study. Cellulose acetate scaffolds were synthesized and kindly provided by Honorary Professor Konstantinos Panagiotou from the Department of Chemical Engineering, Aristotle University of Thessaloniki.

### 2.2. Methods

#### 2.2.1. Scaffold Preparation

Cellulose acetate was dissolved in acetone:dimethylacetamide (DMA) mixtures at a concentration of 20% *w*/*v*. Three different volume ratios of the solvents were used: acetone:DMA 1:0, 1:0.5, and 1:1. In the solutions with a solvent volume ratio acetone:DMA 1:0.5 and 1:1, amoxicillin was dissolved at a concentration of 2% *w*/*v*. First, amoxicillin was dissolved in DMA and then this solution was mixed with acetone to dissolve the cellulose acetate. The distance between the needle (inside diameter 1 mm) and the collector was kept constant at 5 cm at all experiments and the flow rate was also constant at 1 mL/h. At these conditions, the solutions were electrospun at three different voltages (8, 15, 22 kV) [24].

#### 2.2.2. Determination of Amoxicillin Loading

The obtained scaffolds were thoroughly washed with methanol to remove the non-entrapped amoxicillin from the surface. Then, the samples were dried under vacuum overnight. Afterward, they were dissolved in acetone and again dried in order to destroy the encapsulation and to release the amoxicillin. Finally, through the addition of phosphate buffer (pH = 2), amoxicillin was selectively dissolved and these solutions were used for the HPLC measurements. The mobile phase consisted of 20% organic solvent (methanol 20% and acetonitrile 80%) and 80% phosphate buffer (pH = 2) with a flow rate of 1 mL/min. Amoxicillin was eluted isocratically and detected at 230 nm.

#### 2.2.3. *In Vivo* Biotinylation and Purification of SpAD and GFP

The gene of SpAD or GFP with a C-terminal His-tag (6xHis) was cloned to the biotinylation vector pAN5 (Avidity, Denver, CO, USA). This vector encodes a 14-mere sequence (AviTag) that constitutes the minimal peptide substrate that can be biotinylated *in vivo* by the biotin ligase [25,26]. The gene of SpAD or GFP was cloned downstream of the coding sequence for the N-terminal AviTag between the BamHI and HindIII restriction sites. The aforementioned recombinant vector was transformed into the strain AVB101 (Avidity), an *E. coli* B strain that additionally contains a pACYC184 plasmid, namely pBirAcm with a birA gene to overexpress biotin ligase. Expression of both biotin ligase and the AviTag-SpAD or AviTag-GFP fusion construct was induced with isopropyl β-D-thiogalactoside (IPTG) at a final concentration of 1 mM at A_600_ = 0.7. Biotin was also added at the time of induction to a concentration of 50 μM (dissolved in bicine buffer).

Biotinylated SpAD or GFP was purified via its C-terminal His-tag through a Ni-NTA column. Cell pellets carrying the recombinant vector were suspended in lysis buffer [50 mM Tris-HCl [Tris(hydroxymethyl) aminomethane HydroChloride] pH 7.5, 150 mM NaCl, 10% glycerol, 5 mM DTT, and 0.1% *v*/*v* Triton X-100]. Re-suspended cells were treated with low frequency and low energy ultrasound for 20 sand 40 s pause on ice (6 cycles) followed by centrifugation of lysates for 15 min at 15,000× *g* and the supernatant was collected. The supernatant was mixed with an equilibrated column (equilibration buffer 50 mM Tris-HCl pH 7.5 and 150 mM NaCl) for 4 h at 4 °C with rotation. The column was washed three times with wash buffer [50 mM Tris-HCl pH 7.5, 150 mM NaCl and 5 mM imidazole] with rotation for 10 min at 4 °C. Elutions were carried out with elution buffer [50 mM Tris-HCl pH 7.5, 150 mM NaCl, 10% glycerol, 5 mM DTT, and 250 mM imidazole] for 15 min at 4 °C with rotation (three times). Buffer was exchanged with dialysis with 20 mΜ Hepes-KOH, pH = 7.6, 6 mM magnesium acetate, 30 mM ammonium acetate, and 4 mM β-mercaptoethanol (Tico buffer) overnight at 4 °C.

#### 2.2.4. Purification of vWF A1

The gene of vWF A1 was cloned into the pET29c vector, which contains a C-terminal His-tag (6xHis) between the BamHI and HindIII restriction sites. Strain BL21 was transformed with the recombinant pET29c vector and cultured until the absorbance reached A_600_ = 0.6. The expression was induced with IPTG at a final concentration of 1 mM.

The expressed vWF A1 domain was purified via its C-terminal His-tag through a Ni-NTA column according to the manufacturer’s protocol. Cell pellets carrying the recombinant vector were suspended in lysis buffer [50 mM NaH_2_PO_4_, 300 mM NaCl, 10 mM imidazole, lysozyme 1 mg/mL]. Resuspended cells were treated with low frequency and low energy ultrasound for 20 s and 40 s pause on ice (6 cycles) followed by centrifugation of lysates for 15 min at 15,000× *g* and the cell pellet with the inclusion bodies was collected. The cell pellet was resuspended in DNPI-10 [50 mM NaH_2_PO_4_, 300 mM NaCl, 10 mM imidazole, 8 M urea] and dissolved by stirring on ice for 1 h. After centrifugation, the supernatant was mixed with the equilibrated column (equilibration buffer DNPI-10) for 4 h at 4 °C with rotation. The column was washed three times with wash buffer DNPI-20 [50 mM NaH_2_PO_4_, 300 mM NaCl, 20 mM imidazole, 8 M urea] for 10 min at 4 °C with rotation. Elutions were carried out with elution buffer DNPI-250 [50 mM NaH_2_PO_4_, 300 mM NaCl, 2500 mM imidazole, 8 M urea] for 15 min at 4 °C with rotation (three times).

#### 2.2.5. Incubation of *S. aureus* with CA Scaffolds in Human Plasma

The half-life of vWF A1 in human plasma is approximately 12 h, so incubation should start immediately after the blood donation to avoid its degradation. Blood was taken from donors and was centrifuged providing plasma. Plasma from different donors was mixed to equalize the concentration used in all scaffolds.

Human plasma was isolated from 50 mL blood by adding EDTA at a final concentration of 10 mM and centrifuging at 3000× *g* for 15 min. Caution was taken not to disturb the intermediate phase where the white blood cells are present because they can hinder the growth of *S. aureus*. CA scaffolds were placed in microtiter plates and incubated with 3–5 mL of human plasma as well as 500 μL of *S. aureus* (A_600_ = 1.2). Incubation conditions were 3, 6, and 9 h at 37 °C without stirring.

#### 2.2.6. Treatment of Scaffolds for Specific Tethering of SpAD and Incubation of *S. aureus* with Biofunctionalized CA Scaffolds

CA scaffolds were modified according to [10,11] with some modifications. Initially, 5% polyethylenimine (PEI) in sodium carbonate buffer (Na_2_CO_3_) pH 8.2 was applied on the CA scaffolds for 12 h in darkness in order to introduce amine groups (−NH_2_ groups). The exposed −NH_2_ groups were treated as previously described [12,13] with some modifications. A mixture of 50 mg/mL polyethyleneglycol (PEG) (mPEG-NHS ester, MW 5000) and 5 ng/mL biotinylated PEG (Biotin-PEG-NHS ester, MW 3400) was added to the scaffolds (ratio of 1:10^−7^) in the same buffer as above and was left for 3 h in darkness to react with the −NH_2_ groups. The scaffolds were washed with Millipore water and treated with 2 μg/mL streptavidin in Na_2_CO_3_ pH 7.5 for 10 min. Subsequent incubation with 40 nM of biotinylated SpAD in Tico buffer [20 mM Hepes-KOH (pH 7.6 at 0 °C), 6 mM AcMg, 30 mM AcNH4, 4 mM β-mercaptoethanol] took place for 10 min. The non-bound SpAD was washed away with Tico buffer. The overall method for the preparation of scaffolds is depicted in the following figure (Figure 1).

Biofunctionalized or non-scaffolds were transferred to microtiter plates and incubated with 5 mL of LB medium as well as 350 μL of *S. aureus* (A_600_ = 1–1.2) for 6 h at 37 °C. Four different incubation conditions were investigated: *S. aureus* incubated directly with the biofunctionalized cellulose acetate-PEG-SpAD surface, *S. aureus* incubated with the non-biofunctional surface of cellulose acetate (control), *S. aureus* incubated with cellulose acetate scaffold to which non-biotinylated PEG has been added, and *S. aureus* incubated with biofunctionalized cellulose acetate-PEG-SpAD scaffold with the addition of vWF A1.

#### 2.2.7. Sample Treatment for Scanning Electron Microscopy (SEM)

The scaffolds were washed with PBS 1× and placed into a glutaraldehyde solution (2.5% *v*/*v* in Millipore water). After 20 min, they were washed with Millipore water and subsequently immersed into 2 mL of 70% *v*/*v* ethanol for 10 min, into 2 mL of 90% *v*/*v* ethanol for 10 min, and into 2 mL of 100% *v*/*v* ethanol for 10 min. After immersing gradually into the different ethanol solutions, the scaffolds were left to dry overnight and were observed the next day with scanning electron microscopy (SEM).

#### 2.2.8. Molecular Simulations

The docking server zdock of Accelrys (http://zdock.bu.edu/, 15 October 2010) was used in the study to predict the interaction between SpAD and vWF A1. The PROTORP server was also used to study the interface of the SpAD-A1 complex. PDB files, pdb 1SQO.pdb and 1DEE.pdb were used for the vWF A1 domain and SpAD domain, respectively. From the pdf files, we received the regions that are crucial for the interaction.

To identify the 10 most likely interaction complexes, Accelrys’ zdock server was used (www.zdock.com 15 October 2010), and the zdock executable files of the same name (listed below) containing information on the shifts and rotations of the two proteins were extracted. Scripts were extracted in the Perl language, which performs the reconstruction of the highest hierarchical complexes as calculated by the Accelrys server. The structure of SpAD was exported from the original 1DEE.pdb file, downloaded from the pdb.org database. The structure of the vWF A1 domain was also exported from the 1SQO.pdb file. The two aforementioned pdbs were submitted to the docking server zdock. The amino acids between the SpAD and vWF A1 interaction interface were also identified in a special server form. The maximum distance of interaction of the critical amino acids of SpAD with the surface of interaction with vWF A1 was determined to be 6 Å (recommendation according to the instructions of the zdock server). After the calculations, the following files were received: zdock.exe, create_lig, job26816_final_out.txt, lig.job26816.bl.r.pdb (corresponds to SpAD), rec.job26816.bl.r.pdb (corresponds to A1), and create.pl. The job26816_final_out.txt file contained all the information on the hierarchical classification of complexes based on the docking server scoring function and all the translations and rotations required to reconstruct the resulting docking complexes. The script create.pl was executed on Scientific Linux 5. This script reads the data from the job26816_final_out.txt file, calls the executable zdock, and reproduces the provided structures. Then, using the modeling software and the VMD program, the aforementioned complexes were placed in a water sphere with the necessary Na ions to achieve electrical neutrality of the system.

The well-known procedure described above was followed for the solubilization of the obtained complexes in a water sphere 44–48 Å with a distance of the complex boundaries of 10 Å from the sphere boundaries in order to exclude possible problems of interaction of the boundary regions of the sphere with the boundaries of the complex. The ionic neutralization of the system was achieved by appropriate addition of sodium or chloride ions, the minimization of the system for 5000 cycles, and the homogenization of the solvent for 4 ns with the positions of the protein atoms constant. This was followed by a production simulation for 6 to 10 ns at 310 K and under NVT conditions. The temperature was selected to simulate the SpAD and vWF A1 interaction conditions in the human body (37 °C). The criterion for completing the simulation of the interaction of each complex was the stabilization of the value of the interaction energy.

## 3. Results and Discussion

### 3.1. Effect of Solvent Ratio on Fiber Size and Morphology

In Figure 2, the average fiber size of the produced scaffolds (without amoxicillin) is presented for different applied voltages and different solvent ratios. The error illustrated in all figures is the mean average deviation. In Figure 2, SEM pictures of the above-mentioned samples are presented. As can be seen, the addition of DMA in the solvent mixture caused a considerable decrease in the fiber size for all applied voltages. By increasing the voltage, the fiber size appeared to increase (Figure 2). A more careful observation of Figure 1 led to the conclusion that by increasing voltage, the fibers were less uniform (mean average deviation increased). This was visible in the SEM images (Figure 2). The morphology of the fibers, however, remained the same. By increasing the amount of DMA in the solvent mixture (acetone:DMA, 1:1), the morphology of the structure was strongly influenced: beads are produced along with the fibers (Figure 3). These results can be mainly explained by the different volatility of the solutions depending on the DMA amount in the solvent mixture. DMA has a high boiling point (166 °C) in contrast to acetone (56.3 °C). When the solvent system is only acetone (1:0), rapid evaporation occurs during fiber formation. The increase in voltage increases the instability at Taylors cone, which results in the increase in the non-uniformity of the structure (Figure 2 and Figure 3). By adding DMA in the solvent mixture (1:0.5), solvent evaporation is slower, which allows a higher stretching, resulting in the decrease in the fiber size (Figure 2 and Figure 3). By increasing the amount of DMA (1:1) even more, evaporation of the solvent is incomplete even after fiber deposition on the collector, which leads to the formation of beads and coalescence. The fiber coalescence is more obvious as the voltage increases (Figure 3).

### 3.2. Determination of Amoxicillin Loading and Effect of Amoxicillin on Fiber Size and Morphology

In Table 1, the actual loading percentages (mass of amoxicillin per mass of polymer + mass of amoxicillin) for different samples are presented. Of note, electrospinning of the solution with 2% amoxicillin and solvent ratio 1:0.5 at 22 kV was not performed due to the high production of SpArks during the experiment.

In Figure 4, the fiber size for the solvent ratio 1:0.5 and at different applied voltages for samples with and without amoxicillin are illustrated. SEM images for samples with encapsulated amoxicillin are presented in Figure 5. The effect of solvent ratio and applied voltage on the fiber size and morphology was the same as in the samples without amoxicillin. The addition of amoxicillin in the initial solution caused the increase in concentration and viscosity, so some increase in the fiber diameter was expected (Figure 4). The increase in concentration, however, was not enough to overcome the influence of the lower volatility of the solution in the case of the solvent ratio 1:1, and thus the morphology remained the same (beads and coalescence). Considering the loading percentages, they were quite low in all cases and fluctuated between 0.3–1.5 mg of amoxicillin per 1 g of polymer. From a medical point of view, the optimum amount of amoxicillin per g of polymer is relative (depending on the age of the patient, the tissue that is targeted to be regenerated, which defines the rate of polymer degradation, etc.). From a materialistic point of view, it can be considered satisfactory by taking into account that encapsulation was performed along uniform nanofibrous structures (for the solvent ratio 1:0.5).

### 3.3. Purification of SpAD and vWF A1

SpAD protein was expressed in *E. coli*, purified, and *in vivo* biotinylated. Purification of biotinylated SpAD via the 6xHistag is shown in Figure 6A. Purified SpAD protein was detected at ~15 kDa. vWF A1 was expressed in *E. coli* and purified via the 6xHistag, as shown in Figure 6Β. Purified vWF A1 protein was detected at ~20 kDa. Biotinylation of purified proteins was confirmed and analyzed by the DotBlot assay.

### 3.4. Functionalization of Cellulose Acetate Scaffolds with GFP or SpAD

Immobilization of biologically active proteins and enzymes on surfaces remains a key element to produce “smart” surfaces that can be used in a variety of applications such as biosensor chips [27,28,29,30], diagnostic tools and enzyme reactors [31,32,33,34], and protein/peptide microarrays [35,36,37]. Proteins, unlike DNA, are known to be susceptible to loss of activity upon immobilization on surfaces due to unfolding processes [38,39]. Therefore, knowledge of the conformation, orientation, and specific activity of proteins bound to surfaces is crucial for the development and optimization of highly specific and sensitive nano devices. The key element to achieve controlled orientation and conformation is to selectively attach the biomolecule at a predetermined site on the surface. It is also commonly used to fuse proteins with specific tags before the NH_2_-terminal or after the C-terminal region of the studied protein. These additions endow the protein with “extra attachment possibilities” without, however, disturbing its functional conformation.

One such biofunctionalization method is via the streptavidin-biotin bond. Streptavidin is a tetrameric protein with four biotin binding sites, which is structurally similar to avidin. The interaction between streptavidin and biotin is one of the strongest non-covalent bonds in nature (Ka = 10^15^ instead of 10^7^–10^11^ for antibody/antigen interaction). The strength of the bond as well as the small size of biotin (MW = 244.3) ensures an ideal system for affinity binding with numerous applications. Because streptavidin lacks any carbohydrate modification and has a near-neutral pI, it has the advantage of much lower non-specific binding than avidin.

CA scaffolds are known to have some non-adherent properties against *S. aureus* [40] and they were initially treated with PEI solution to introduce −NH_2_ groups on the surface. The exposed −NH_2_ groups were used to covalently link a layer of PEG on top, which was sparsely biotinylated. Streptavidin was attached specifically on the biotin of the PEG layer. Since streptavidin has four binding positions for biotin, after attachment on the surface, it still has three free positions to bind a second biotin. Thus, biotinylated SpAD was immobilized on the scaffolds after binding the streptavidin of the surface.

In order to check the success of the procedure, biotinylated GFP was also immobilized under the same conditions. Subsequently, the CA scaffolds with GFP (CA-GFP) were studied under a confocal fluorescence microscope (Nikon Eclipse), as shown in Figure 7.

The advantage of GFP lies in the fact that only active molecules that acquire their final conformation retain their fluorescence. The successful immobilization of GFP on the surface verifies the biofunctionalization of the scaffolds as well as the successful immobilization of SpAD indirectly. Additionally, after immobilization of GFP, the micromorphology of the polymeric substrate could be clearly seen.

### 3.5. Incubation of S. aureus or Ε. coli with CA Scaffolds

*S. aureus* was incubated with CA scaffolds in the presence of complete growth medium (LB) for 24 and 72 h. According to previous studies, CA has a great antibacterial potential particularly against *S. aureus*. The results show that *S. aureus* can form colonies on the CA scaffold after 24 h of incubation (Figure 8).

Similarly, *E. coli* was incubated with CA scaffolds in the presence of complete growth medium (LB) for 24 and 72 h. The results showed that CA scaffolds compromise an unfavorable material for the growth of *E. coli*. Figure 9 shows the only areas that *E. coli* colonies were observed, while on the rest of the scaffold surface, the growth was almost zero.

The sensitivity of *E. coli* against amoxicillin is clearly shown in Figure 9a,b. In contrast, the growth of *S. aureus* on the same scaffolds and under the same time periods (24 h and 72 h) was not significantly deteriorated or at all inhibited (Figure 8a,b).

### 3.6. Incubation of S. aureus with CA Scaffolds in Human Plasma

*S. aureus* cultures were incubated for 3, 6, and 9 h with CA scaffolds in the presence of human plasma. The results showed that human plasma is an excellent culture material for *S. aureus* (Figure 10). The local development of *S. aureus* was evident with areas of intense, medium, and low growth. The growth density of *S. aureus* did not differ significantly between 3 and 6 h, however, the colonies formed were somewhat larger. After 9 h of incubation, the areas that are no longer covered with attached *S. aureus* colonies were minimal. It is also surprising that *S. aureus* colonizes deep down in the scaffold’s fibers. According to Cardile et al. (2014), human plasma enhances the expression of proteins responsible for the *S. aureus* attachment to the matrix and alters the microbial tolerability [41]. These results are in accordance with our results, which show that colonies are formed at a high rate and the antimicrobial properties of cellulose acetate are altered.

### 3.7. Incubation of S. aureus with the Biofunctionalized CA Scaffolds

In order to evaluate the impact of vWF A1 addition on *S. aureus* growth, three biofunctionalized CA scaffolds were incubated with an overnight culture of *S. aureus* in LB medium. The first scaffold only had a PEG layer SpArsely biotinylated (CA-PEG), the second had SpAD immobilized on the PEG layer via a biotin-streptavidin bond (CA-SpAD), and the third was similar to the second, but in addition, the culture of *S. aureus* contained vWF A1 (CA-SpAD/vWF A1) with a final concentration of ~80 μg/mL. All three scaffolds were incubated with *S. aureus* (or the *S. aureus* with the vWF A1 in the third case) for 6 h at 37 °C. The three scaffolds were studied with SEM and they were also compared to a non-functionalized CA scaffold (CA-control) (Figure 11).

The results show that *S. aureus* grew in all four scaffolds. There were areas on the same scaffold where the growth was stronger and other areas where the growth was minimal or zero. Nevertheless, there were important differences among the four scaffolds in the size of the *S. aureus* colonies as well as in the magnitude of the surface coverage by the colonies. The non-functionalized CA scaffold, in contrast to the literature data [41] and despite the fact that it contained the antibiotic amoxicillin, exhibited a major interaction with *S. aureus*, which was substantially grown on it and presented the highest surface coverage compared to the functionalized scaffolds.

In both the CA-PEG and CA-SpAD scaffolds, the attachment of *S. aureus* was significantly reduced compared to the CA-control. The CA-PEG scaffold showed a minor growth of *S. aureus* colonies, leading to the conclusion that PEG has non-adherent properties for *S. aureus*. The CA-SpAD scaffold also exhibited non-adherent properties; nevertheless, it also showed some minor growth of *S. aureus* colonies. Finally, the CA-SpAD/vWF A1 showed the least surface coverage among all the scaffolds.

### 3.8. Incubation of S. aureus with Biofunctionalized CA Scaffolds in Human Plasma

As shown in the schematic illustration when SpAD is immobilized on the scaffold, *S. aureus* cannot attach on the surface since it does not contain binding sites for its own protein (Figure 12a). In the case where soluble vWF A1 is also present, the binding sites of the immobilized SpAD are occupied as well as the binding sites of vWF A1, which are able to bind the *S. aureus* already occupied by the immobilized SpAD. In this way, the surface has no available binding positions for *S. aureus,* thus growth of the bacterium and the potential infection are hindered (Figure 12b). The possibility is also high that the vWF A1 is binding to *S. aureus* during the incubation, since their interaction is quite favorable, as mentioned in the literature [42] (Figure 12 inset). However, in this case, all the binding sites were also occupied and the bacterium growth on the surface was again hindered (Figure 12c).

The first explanation has to take into account the fact that the concentration of *S. aureus* in the case of bacteremia or septicemia is between 10 and 100 colony forming units/mL (CFU/mL) [42]. In this study, considering that the CA-SpAD scaffold was incubated with an overnight culture and the exponential growth of *S. aureus* during the incubation in the microtiter plate could not be avoided, the concentration was expected to be higher. Therefore, it is not astonishing that the non-adherent properties of the scaffold were worse than in a potential *in vivo* use of the same scaffold in the plasma. A methodology that could ensure that the concentration of *S. aureus* in the microtiter plate is similar to the one during bacteremia would lead to more precise conclusions on the ability of the biofunctionalized scaffold to inhibit the *in vivo* attachment of *S. aureus*. Another explanation could be the insufficient knowledge of the activity of more than 50 known MSCRAMMs of *S. aureus*. Thus, a potential interaction between one of these proteins and SpAD, leading to a partial annulment of the non-adherence of the scaffold, cannot be excluded.

Nevertheless, the main purpose of this study was to hinder the attachment of *S. aureus* on surfaces in the plasma, where the immobilized SpAD interacts with the vWF A1 domain. According to the results, the CA-SpAD/vWF A1 scaffold presents exceptional non-adherent properties. Despite this, further studies are required to clarify the mechanism that allows *S. aureus* to partially bypass the non-adherent properties of CA-SpAD when vWF A1 is not present.

### 3.9. Interaction between SpAD and vWF A1

The exact region with which SpAD binds to the A1 domain has been accurately characterized by the critical amino acid mutations of the SpAD region involved in the interaction [41]. Crystallographic resolution of the structure of the SpAD protein complex and the Fab-VH region of the IgM immunoglobulin [42] with which SpAD is known to react, contributed significantly to the determination of the correct amino acids involved in this interaction. The above work demonstrated that the region with which SpAD interacts with the vWF A1 domain is the same with the region with which it interacts with IgM. Specifically, α-helices A2 and A3 of SpAD participate in the interface of interaction with the A1 domain, and in particular, the following amino acids gln2808, phe2811, tyr2812, leu2815, arg2825, asn2826, phe2828, ile2829, lys2833 are crucial for this interaction. The following experiments involving docking studies and molecular dynamics simulations were used to determine the exact position of the vWF A1 to which SpAD is attached.

The first 10 structures that resulted from the execution of the script were used. Illustration of the first 10 structures quickly makes clear that there are three families of structures. This means that each family of structures has many similarities in the interaction position of vWF A1, while the position of SpAD has minor differences regarding its position (translation-rotation) and its connection angle with A1. It also became apparent that the binding site on vWF A1 had many similarities to the binding site of SpAD to the Fab subunit of IgM. Two opposite β-leaves (β2 and β3 in the case of the A1 domain) were located opposite the nine critical amino acids to which SpAD binds to the vWF A1 in both cases.

Binding energies were calculated for the first six complexes and the results showed the initial binding energy decreased rapidly for structures 1, 2, 3, 4, and 5 (see Appendix A). The structure with the lowest connection energy (i.e., structure 6 maintained a very low binding energy (−370 kcal/mol).

A more detailed description of the interface of complex 6 SpAD-A1 was obtained using the PROTORP server (http://www.bioinformatics.sussex.ac.uk/protorp/, 25 October 2010). The pdb was submitted to this server, from which the following results were received. The area of the interface accessible surface area was 429.29 Å2, which was 4.39% of the total area of the complex. A total of 38 atoms participated in the interface, of which 42.3% were polar, 18.6% were non-polar, and 37.9% were neutral. The amino acid residues present at the interface were 16, for which the ratios of 43.7% (polar), 25% (non-polar), and 18.75% (charged) applied, respectively. The corresponding ratio of polar amino acids to the total surface of the complex was 32.22% (much lower than the corresponding ratio at the interface) and the non-polar was 32.21%, a finding that shows that the interaction of SpAD and vWF A1 is mainly electrostatic in nature. Additionally, the results showed that the interaction of the two domains was enhanced by six hydrogen bonds and 11 salt bridges between amino acid residues at the interface. Figure 13 shows the interaction of SpAD and vWF A1.

As observed, the amino acids of SpAD that participate in the SpAD-A1 interface were very similar to those that were identified experimentally by O’Seagdha et al. [42] for SpAD-IgM interaction. The nine crucial amino acids, gln2808, phe2811, tyr2812, leu2815, arg2825, asn2826, phe2828, ile2829, lys2833 were located opposite the two opposite β-leaves (β2 and β3 in the case of the A1 domain) and the distance of the involved amino acids of A1 from SpAD was 6 Å, as required by the zdock algorithm to determine the interaction surface.

## 4. Conclusions

This study makes it clear that the immobilization of appropriate biomolecules on surfaces could be an effective novel strategy to minimize the spread of hospital acquired infections caused by *S. aureus.* Cellulose acetate scaffolds that inhibit *S. aureus* growth were successfully produced via electrospinning. Concerning their production method, the increase in DMA in the solvent system had a tremendous influence on the fiber size and morphology, which can either decrease the fiber diameter or lead to bead formation. The increase in voltage resulted in not many uniform structures. Encapsulation of amoxicillin is feasible with low loading percentage, independent of the solvent ratio used or the applied voltage. The addition of amoxicillin in the initial solution results in a small increase in the fiber size but has no influence on the morphology. Concerning the treatment of the scaffolds, the immobilization of SpAD led to surfaces with non-adherent properties. These properties were further enhanced from the interaction with the vWF A1. As shown in the molecular simulation experiments, SpAD binds to vWF A1 and nine amino acids are crucial for this interaction, the same as for the interaction with IgΜ.

The improved surface properties generated from vWF A1 binding to SpAD make the CA-SpAD/vWF A1 scaffold very useful for covering surfaces of biomedical interest that come into contact with human blood such as medical equipment, clothing, and gloves as well as implants and stents. Still, important steps should be made toward scaling up the production of biofunctionalized scaffolds in order to expand the armory of modern medicine against the pathogens that continue to inflict mankind.

## Figures and Tables

**Figure 1 jfb-13-00021-f001:**
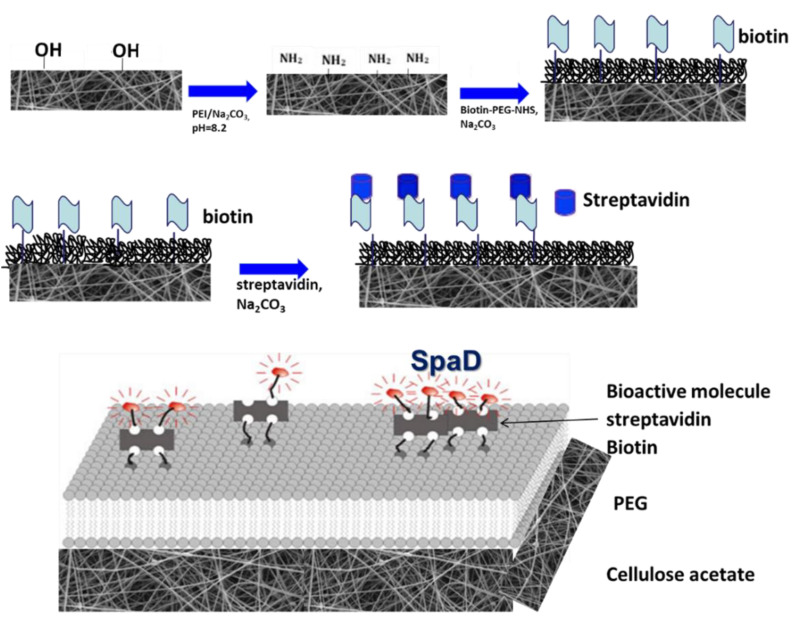
Representation of the overall procedure concerning the preparation and treatment of scaffolds for specific tethering of SpAD.

**Figure 2 jfb-13-00021-f002:**
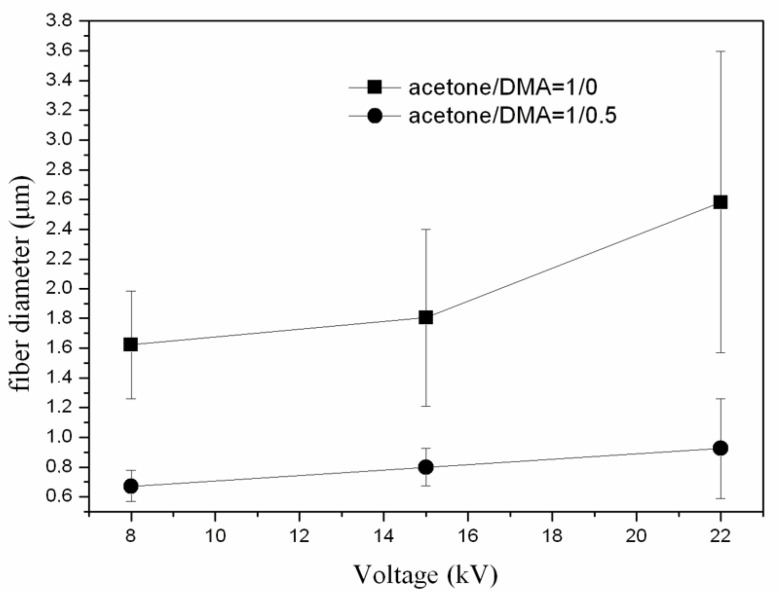
Fiber size as a function of applied voltage for different solvent ratios.

**Figure 3 jfb-13-00021-f003:**
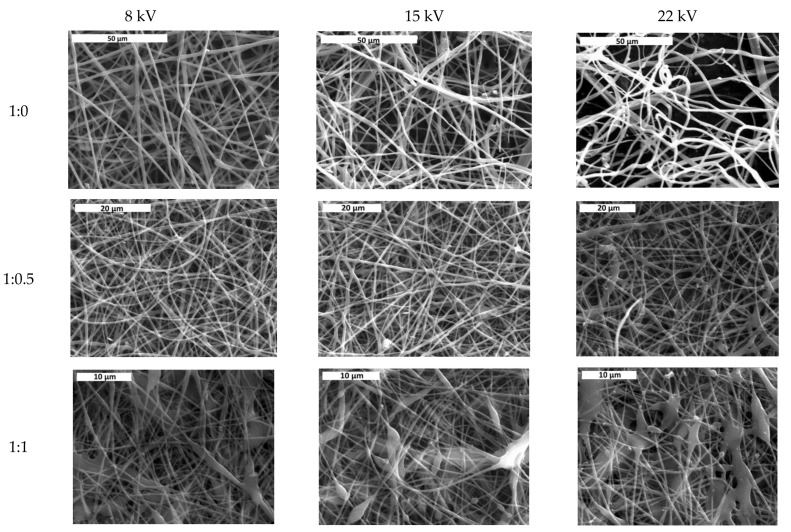
SEM images of samples prepared at different solvent ratios and applied voltages. Scale bar is 50, 20, and 10 μm for solvent ratio 1:0, 1:0.5, and 1:1, respectively.

**Figure 4 jfb-13-00021-f004:**
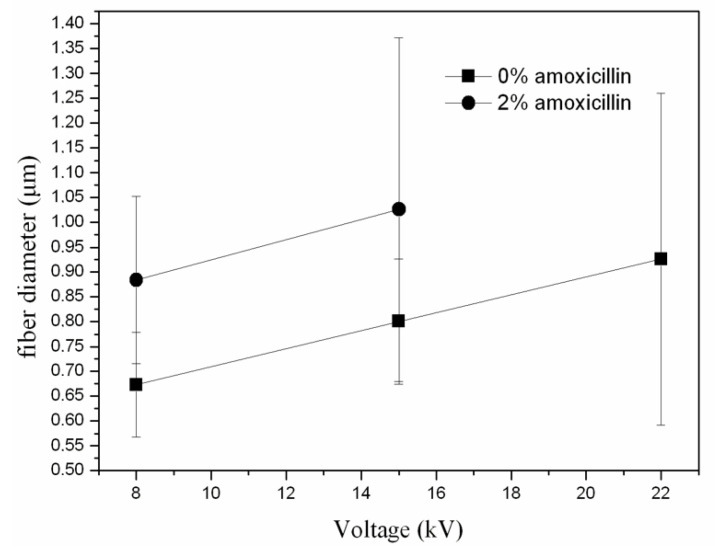
Fiber size as a function of applied voltage for samples with and without amoxicillin.

**Figure 5 jfb-13-00021-f005:**
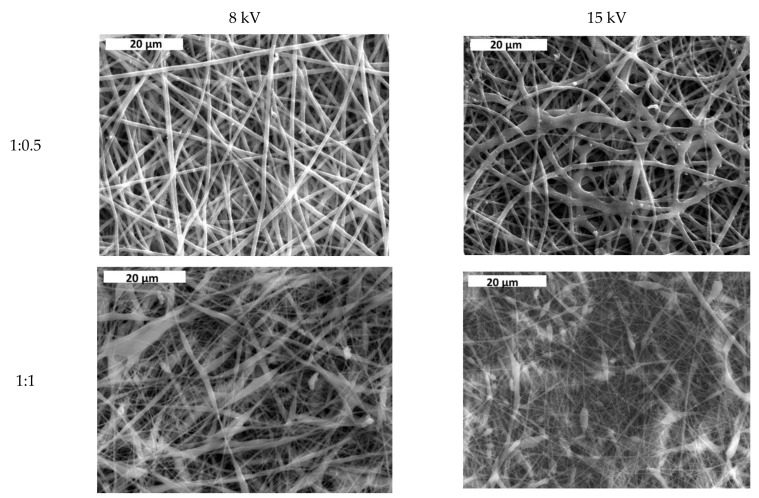
SEM images of samples with encapsulated amoxicillin.

**Figure 6 jfb-13-00021-f006:**
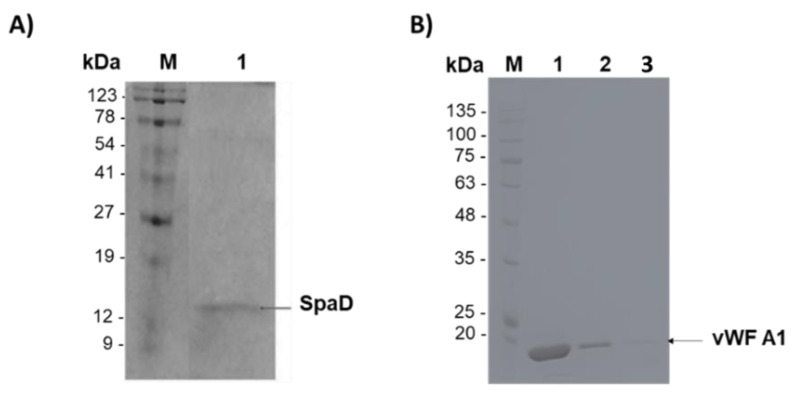
12% *v*/*v* SDS-PAGE electrophoresis for the purification of (**A**) SpAD. M: protein markers, (1) first elution after dialysis. The arrows show the entire protein at ~15 kDa, (**B**) vWF A1, M: protein markers, (1) first elution after dialysis, (2) second elution after dialysis, (3) third elution after dialysis. The arrows show the entire protein at 20 kDa.

**Figure 7 jfb-13-00021-f007:**
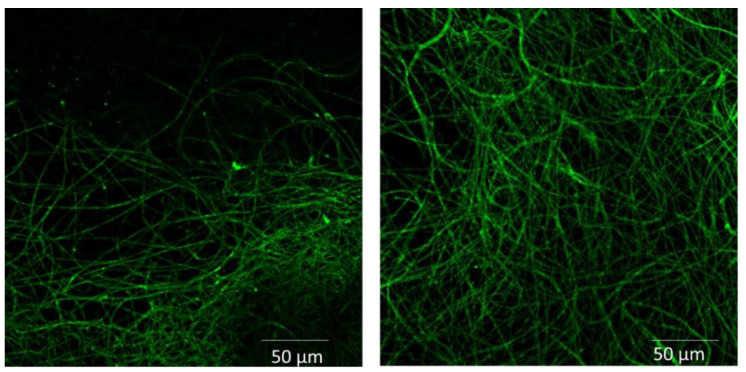
Biofunctionalization of the CA scaffold with GFP. Photos taken with a confocal fluorescence microscope.

**Figure 8 jfb-13-00021-f008:**
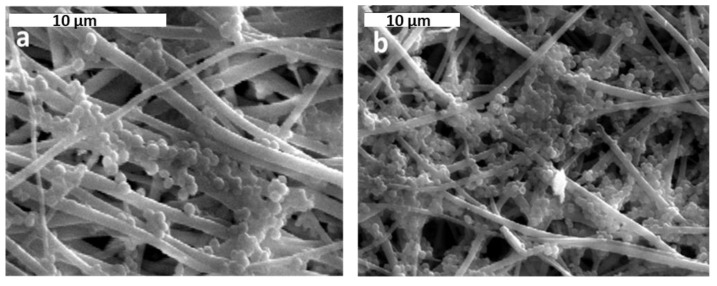
Comparative juxtaposition of *S. aureus* growth on CA scaffolds with LB medium. (**a**) After 24 h and (**b**) 72 h.

**Figure 9 jfb-13-00021-f009:**
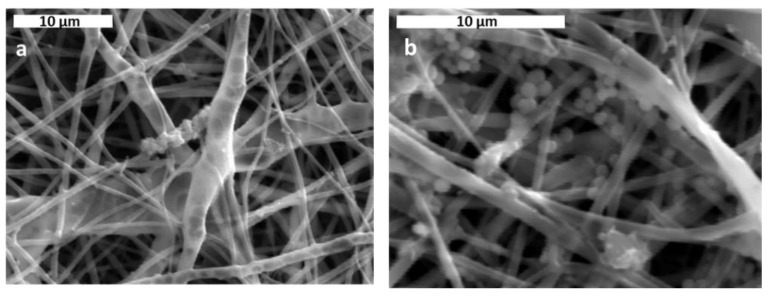
Comparative juxtaposition of *E. coli* growth on CA scaffolds with LB medium (**a**) after 24 h and (**b**) after 72 h.

**Figure 10 jfb-13-00021-f010:**
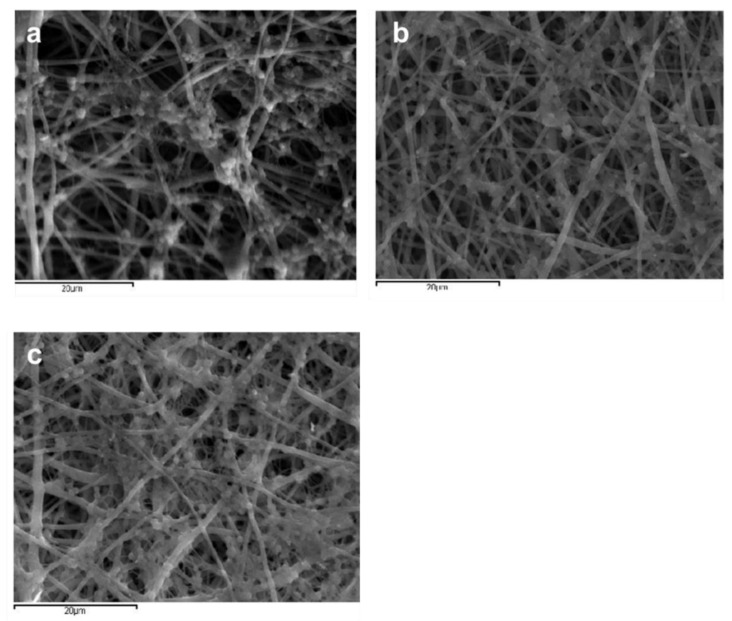
Comparative juxtaposition of *S. aureus* growth on CA scaffolds with human plasma. (**a**) After 3 h, (**b**) after 6 h, (**c**) after 9 h.

**Figure 11 jfb-13-00021-f011:**
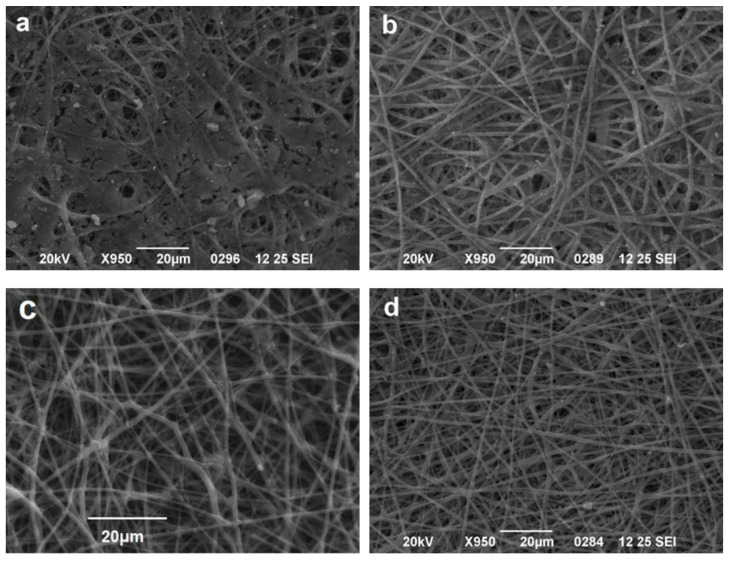
Comparative juxtaposition of *S. aureus* growth on CA scaffolds. (**a**) Non-functionalized scaffold (CA-control), (**b**) with PEG layer (CA-PEG), (**c**) with immobilized SpAD on the PEG layer (CA-SpAD), and (**d**) with immobilized SpAD on the PEG layer in the presence of vWF A1 (CA-SpAD/vWF A1).

**Figure 12 jfb-13-00021-f012:**
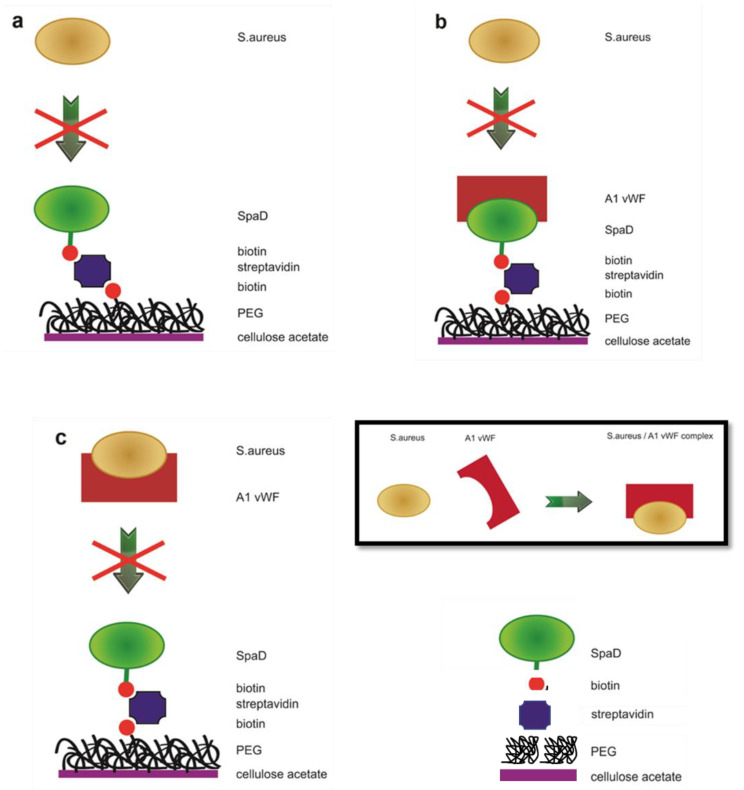
Schematic illustration of the final biofunctionalized surface and inhibition of *S. aureus* attachment. (**a**) *S. aureus* does not attach to the CA-SpAD surface because it does not have binding sites for its own protein, (**b**) *S. aureus* does not attach to the CA-SpAD/vWF A1 surface because the binding sites of vWF A1 and SpAD are already occupied, (**c**) *S. aureus* does not attach to the CA-SpAD/vWF A1 surface because the binding sites of vWF A1 and *S. aureus* are already occupied, inset *S. aureus* binds to the vWF A1 with high affinity in physiological conditions.

**Figure 13 jfb-13-00021-f013:**
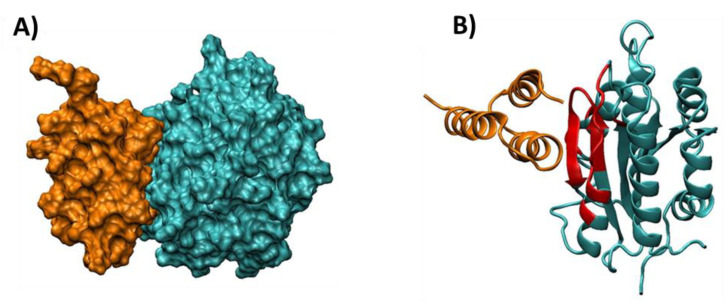
Illustration of the interaction of the A1 region and SpAD (complex 6). (**A**) Surface illustration cyan: A1, orange: SpAD. (**B**) Schematic illustration of the vWF A1 amino acids involved in the interaction interface of the A1 and SpAD domains. The crucial amino acids for the interaction mentioned are shown in red and belong to the two opposite β-sheets (β2 and β3), the loop and the α-helix of the vWF A1 domain that face the A2 and A3 helixes of SpAD. Orange: SpAD, Cyan: vWF A1.

**Table 1 jfb-13-00021-t001:** Loading percentages of amoxicillin onto cellulose acetate scaffolds (g of amoxicillin/100 g of amoxicillin + polymer).

	7 kV	15 kV	22 kV
1:0.5	0.07	0.06	-
1:1	0.06	0.16	0.03

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
