# Peer review of "SpAD Biofunctionalized Cellulose Acetate Scaffolds Inhibit Staphylococcus aureus Adherence in a Coordinating Function with the von Willebrand A1 Domain (vWF A1)"

_jfb, 2022, doi:10.3390/jfb13010021_

Round 1
Reviewer 1 Report
The manuscript is well written and described with proper experimentation results. It can be recommended for acceptance after some revision as suggested below:
,
- grammatical errors and typos are very prevalent all over the manuscript. It should be corrected.
- The title is too much lengthy. it must be changed for a better readability.
- The hypothesis of the work is missing in introduction. Although author is focused on streptococcus however they should also mention about pathogenicity of other pathogens also that can be relevant to use the technique.
- A pictorial presentation of the over all method for scaffold preparation will be better.
- discussion of bioinformatics data has not been integrated with the story properly. I will suggest to include a final paragraph in the discussion section with an integration of overall oberservations.
- recent citaions can be included like https://doi.org/10.1007/s10565-021-09587-z,https://doi.org/10.1080/21691401.2018.1503598 , https://doi.org/10.1021/acs.jproteome.6b00983
Reviewer 2 Report
jfb-1510673 fabricated a cellulose acetate based scaffolds with non-adherent property for Staphylococcus aureus. The topic and results are interesting. The title is too long and should be shortened. The abstract did not contain valuable experimental results. L32-36: The detriments of Staphylococcus aureus are not introduced in detail. Since the scaffolds are designed for treating Staphylococcus aureus, the content related with E. coli (L69) should be deleted. In most cases, the authors only described the results without thoroughly discussing them. Throughout the manuscript, the authors should use more specific quantitative index to evaluate the antibacterial effect of the scaffolds, avoiding some descriptive sentences such as L400. The scale bars in the figures are too small. Results in the tables should be presented as mean ± SD. In figure 2, which is the meaning of 1/0, 1/0/5 and 1/1 is not clear. The quality of SDS-PAGE electrophoresis result in figure 5 is poor. In figure 11, the meaning of each geometrical pattern is not clear. L442: “Interaction”. The format of reference list is not very standard.
Author Response
Please see the attachmen

Reviewer 3 Report
Dear Editor and Authors,
I read carefully the article and I can conclude that is an interesting paper, but some issues must be clarified:
- Please define DMA in the paper.
- Please for the SEM images the scale bar is noisy.
- In figure 6 add the scale bar.
- Check the English.
Round 2
Reviewer 2 Report
Accept in present form